# Monocyte MRI Relaxation Rates Are Regulated by Extracellular Iron and Hepcidin

**DOI:** 10.3390/ijms24044036

**Published:** 2023-02-17

**Authors:** Praveen S. B. Dassanayake, Rahil Prajapati, Neil Gelman, R. Terry Thompson, Frank S. Prato, Donna E. Goldhawk

**Affiliations:** 1Imaging Program, Lawson Health Research Institute, London, ON N6A 4V2, Canada; 2Medical Biophysics, Western University, London, ON N6A 5C1, Canada; 3Collaborative Graduate Program in Molecular Imaging, Western University, London, ON N6A 5C1, Canada

**Keywords:** magnetic resonance imaging, monocytes, iron export, ferroportin, hepcidin, inflammation

## Abstract

Many chronic inflammatory conditions are mediated by an increase in the number of monocytes in peripheral circulation, differentiation of monocytes to macrophages, and different macrophage subpopulations during pro- and anti-inflammatory stages of tissue injury. When hepcidin secretion is stimulated during inflammation, the iron export protein ferroportin is targeted for degradation on a limited number of cell types, including monocytes and macrophages. Such changes in monocyte iron metabolism raise the possibility of non-invasively tracking the activity of these immune cells using magnetic resonance imaging (MRI). We hypothesized that hepcidin-mediated changes in monocyte iron regulation influence both cellular iron content and MRI relaxation rates. In response to varying conditions of extracellular iron supplementation, ferroportin protein levels in human THP-1 monocytes decreased two- to eightfold, consistent with paracrine/autocrine regulation of iron export. Following hepcidin treatment, ferroportin protein levels further decreased two- to fourfold. This was accompanied by an approximately twofold increase in total transverse relaxation rate, R_2_*, compared to non-supplemented cells. A positive correlation between total cellular iron content and R_2_* improved from moderate to strong in the presence of hepcidin. These findings suggest that hepcidin-mediated changes detected in monocytes using MRI could be valuable for in vivo cell tracking of inflammatory responses.

## 1. Introduction

During an inflammatory response, circulating peripheral blood monocytes are recruited to the inflammation site by receptor-mediated interactions with chemokines. This monocyte activation leads to interactions with cell adhesion molecules on the activated endothelium and extravasation into the target tissue [1]. Moreover, an increase in monocyte-related chemokines during inflammation leads to monocyte proliferation, directly correlating with an increase in the total number of monocytes [1]. Due to these inflammation-related chemokines, monocytes are one of the major immune cell types involved in chronic inflammatory conditions such as atherosclerosis and heart failure. Furthermore, an increase in different monocyte sub-populations during pro- and anti-inflammatory stages may serve as markers for monitoring chronic inflammatory conditions [2,3].

During an inflammatory response, monocytes play a vital role in host defense mechanisms against pathogens by limiting the availability of iron required for bacterial growth. This is facilitated by upregulation of the endocrine hormone hepcidin [4]. During inflammation, hepcidin binds to the iron export protein ferroportin (FPN) and induces its internalization and degradation, thereby increasing iron retention in monocytes and macrophages [5]. Monocytes are the precursors of macrophages, which can be divided into at least two categories: M1 (pro-inflammatory) and M2 (anti-inflammatory) macrophages. While there is a spectrum of macrophage phenotypes, in general, M1 displays an iron storage phenotype while M2 exhibits an iron recycling phenotype [6]. These features of iron metabolism may allow us to differentiate between pro- and anti-inflammatory stages using magnetic resonance imaging (MRI).

MRI is a noninvasive imaging modality that has been used to track cellular activities during inflammation [7,8]. Iron-based contrast agents such as superparamagnetic iron oxide particles have been developed to enhance image contrast and improve the tracking of inflammatory cells [7]. Since monocytes and macrophages are phagocytes, they internalize these exogenous contrast-enhancing particles, thereby promoting more efficient tracking of these cells using MRI [7]. However, apart from phagocytic activity, the effect of inflammation-associated changes in iron metabolism on the cellular magnetic resonance (MR) signal have not been reported. An endogenous biomarker of monocyte activity would be ideal not only for long-term monitoring but also for reporting tissue-specific changes in iron metabolism in real time. Therefore, given the role of hepcidin in regulating iron export and the role of monocytes in inflammation, we hypothesized that MRI may have potential for tracking changes in iron homeostasis related to inflammatory responses. To test this, we used the human monocyte THP-1 cell line to examine endogenous cellular MR signals resulting from varying extracellular iron and hepcidin to mimic the response to tissue injury (hemorrhage) and pro-inflammatory signaling [9], respectively.

## 2. Results

### 2.1. Effect of Iron and Hepcidin on FPN Expression

To obtain evidence of iron export activity in THP-1 monocytes, cultured cells were incubated in the presence and absence of iron supplementation. Western blots revealed FPN expression in all samples (Figure 1A). Under standard culture conditions (−Fe), THP-1 cells express maximal levels of iron export protein, similar to the level of GAPDH expression (ratio ~1 in Figure 1C, blue bar). Moreover, after 7 days of iron supplementation (+Fe), there was little or no change in the expression of FPN, as confirmed by densitometry (Figure 1C, blue bars). However, up to two hours after the withdrawal of iron supplement, FPN expression significantly declines (+Fe versus 1 h-Fe, *p* < 0.01; +Fe versus 2 h-Fe, *p* < 0.01), before significantly recovering at 4 h after the withdrawal of extracellular iron (2 h-Fe versus 4 h-Fe, *p* < 0.05; Figure 1A,C, blue bars). After 24 h of iron supplement withdrawal (24 h-Fe), FPN expression significantly decreased (+Fe versus 24 h-Fe and 4 h-Fe versus 24 h-Fe, *p* < 0.001).

To determine the effect of the polypeptide hormone hepcidin on FPN expression, cells were treated with exogenous hepcidin (Figure 1B). Potential changes in FPN expression in response to hepcidin were assessed using western blotting to track protein degradation. One-way ANOVA suggests that FPN expression significantly decreases (Figure 1C, orange bars) at 4 and 24 h after iron supplement withdrawal in hepcidin-treated samples (+Fe orange versus 4 h-Fe orange, *p* < 0.01; +Fe orange verses 24 h-Fe orange, *p* < 0.05).

The two-way ANOVA comparing hepcidin treated and untreated samples (red lines) indicates that at baseline (−Fe), FPN expression was significantly decreased in response to hepcidin (−Fe blue verses −Fe orange, *p* < 0.001), confirming a biologically meaningful interaction. Likewise, in the presence of iron supplementation (+Fe) and exogenous hepcidin, FPN protein levels were significantly decreased (+Fe blue versus +Fe orange, *p* < 0.001). Interestingly, one hour after the removal of iron supplement and hepcidin treatment (1 h-Fe), FPN is still expressed at a low level, suggesting little influence of exogenous hepcidin. This decreases even further after 4 h of iron supplement withdrawal (4 h-Fe blue versus 4 h-Fe orange, *p* < 0.001) suggesting that FPN expression is down-regulated in the presence of hepcidin at these timepoints.

In co-culture models, proinflammatory cytokines such as interleukin 6 are reported to stimulate hepcidin expression in one cell type (e.g., hepatocytes) to promote endocrine responses in circulating leukocytes [10,11]. To build on these cell culture models of hepcidin activity, we examined the response of THP-1 monocytes to hepcidin secreted by multipotent P19 cells: an iron-exporting, mouse embryonic carcinoma, suspected of hepcidin production for paracrine/autocrine regulation of FPN [12]. To validate both the response of THP-1 cells to secreted hepcidin and the production of functional hormone by P19 cells, we cultured monocytes in P19 cell-conditioned medium (presumed to contain secreted hepcidin upon stimulation) and examined the effect on FPN expression in monocytes. Compared to the control media, there was a significant decrease in FPN protein levels when THP-1 cells were exposed to stimulated conditioned medium (Figure 2 and Appendix A). These data not only support the evidence for an endocrine response of THP-1 monocytes to hepcidin, but also validate the P19 cell model of hepcidin-regulated iron export.

### 2.2. Effect of Hepcidin on Cellular Iron Content

Despite changes in FPN expression, cellular iron content does not change significantly across all time points, regardless of changes in iron supplementation (Figure 3A). There was no correlation between the ratio of FPN/GAPDH and total cellular iron content. Elemental iron analysis using ICP-MS shows that cellular iron content ranges between 0.175–0.679 µg/mg protein, indicating a modest rise in iron content and return to the baseline. However, in the presence of iron supplementation and hepcidin (+Fe) there is a significant increase (*p* < 0.01) in total cellular iron content (Figure 3B) compared to the baseline control (−Fe versus +Fe, *p* < 0.01). Subsequently, one hour after the withdrawal of iron supplement (1 h-Fe), hepcidin treatment results in a significant decrease in total cellular iron content (+Fe versus 1 h-Fe, *p* < 0.05, Figure 3B). Interestingly, two hours after the removal of iron supplement (2 h-Fe), hepcidin treatment results in a second increase in total cellular iron content (1 h-Fe versus 2 h-Fe, *p* < 0.001) which is sustained up to 4 h-Fe and then followed by a significant decrease in cellular iron content (4 h-Fe versus 24 h-Fe, *p* < 0.01). These data reflect a biphasic response of THP-1 cells to hormonal treatment with hepcidin.

### 2.3. Effect of Iron Supplement on Transverse Relaxation

To examine the effect of changes in extracellular iron on transverse relaxation, THP-1 monocytes were mounted in an MR phantom and scanned at 3T. The mean transverse relaxation rate of these cells under different conditions of iron treatment indicates a relatively high signal (approximately > 10 s^−1^) for both the total transverse relaxation rate (R_2_*, Figure 4A) and its irreversible component (R_2_, Figure 4B). Neither R_2_* nor R_2_ was influenced by continuous iron supplementation (+Fe) compared to the control culture (−Fe). However, withdrawal of iron supplementation for 1 (1 h-Fe) to 4 (4 h-Fe) hours significantly increased the R_2_* signal (+Fe versus 1 h-Fe, *p* < 0.01; +Fe versus 2 h-Fe, *p* < 0.001; +Fe versus 4 h-Fe, *p* < 0.01). However, by 24 h (24 h-Fe), R_2_* returned to baseline values (4 h-Fe versus 24 h-Fe, *p* < 0.01; Figure 4A).

The effect of iron supplement withdrawal on R_2_ was not apparent for 4 h (4 h-Fe) at which point R_2_ significantly increased over baseline values (−Fe versus 4 h-Fe, *p* < 0.05) before returning to control levels (4 h-Fe versus 24 h-Fe, *p* < 0.01).

The R_2_* signal is made up of two components: reversible R_2_’ and irreversible R_2_. Consistent with R_2_* and R_2_, the R_2_’ signal (Figure 4C) showed no significant change in the presence of iron supplementation (+Fe) compared to the control culture (−Fe). However, R_2_’ significantly increased 1 (1 h-Fe) to 4 (4 h-Fe) hours after iron supplementation was withdrawn (+Fe versus 1 h-Fe, *p* < 0.01; +Fe versus 2 h-Fe, *p* < 0.001; +Fe versus 4 h-Fe, *p* < 0.05). By 24 h after the removal of iron supplement (24 h-Fe), R_2_’ had returned to baseline values (4 h-Fe versus 24 h-Fe, *p* < 0.001).

### 2.4. Effect of Hepcidin on Transverse Relaxation

THP-1 cells were incubated with various amounts of iron supplement and hepcidin before scanning at 3T to measure relaxation rates. The mean values of R_2_*, R_2_ and R_2_’ are shown in Figure 5. By adding hepcidin, we aimed to interrupt the iron export activity of monocytes. Total transverse relaxation rate (R_2_*) significantly increased (*p* < 0.05) in the presence of iron supplementation and hepcidin (−Fe versus +Fe, Figure 5A) and remained high for 1 h after withdrawing iron supplementation (+Fe versus 1 h-Fe). However, 2 h after iron supplementation withdrawal (2 h-Fe) R_2_* significantly increased again (1 h-Fe versus 2 h-Fe, *p* < 0.001) and remained high until 4 h after iron supplement withdrawal, when the signal returned to baseline (4 h-Fe versus 24 h-Fe, *p* < 0.001).

Under the same conditions, the irreversible component (R_2_, Figure 5B) showed significant changes similar to R_2_*. The reversible component (R_2_’, Figure 5C) remained unchanged in the presence of iron supplement (+Fe) compared to the baseline (−Fe). However, R_2_’ significantly increased (*p* < 0.01) after 2 h of iron withdrawal (−Fe versus 2 h-Fe, *p* < 0.01; 1 h-Fe versus 2 h-Fe, *p* < 0.01) before returning to baseline (2 h-Fe versus 24 h-Fe, *p* < 0.01).

### 2.5. Comparison of Transverse Relaxation between Hepcidin Treated and Untreated THP-1 Cells

The influence of hepcidin on THP-1 samples was evaluated using two-way ANOVA. Figure 6 indicates that in the presence of both hepcidin and various levels of extracellular iron, there are significant changes in relaxation rates. R_2_* (Figure 6A) significantly increases (*p* < 0.05) in the presence of both extracellular iron and hepcidin (+Fe, orange bar) compared to monocytes treated with iron alone (+Fe, blue bar). R_2_* also significantly decreases within 1 h of iron supplement withdrawal (1 h-Fe) in the presence of hepcidin (orange bar versus blue bar). There was no influence of exogenous hepcidin on R_2_* at any other time point. Likewise, R_2_ was not influenced by hepcidin (Figure 6B) under any of the conditions examined. A comparison between R_2_’ (Figure 6C) signals indicates that hepcidin predominantly influences the reversible component of transverse relaxation rate.

### 2.6. Influence of Iron Supplement and Hepcidin on R_1_ Longitudinal Relaxation

The R_1_ signal remained constant across all treatment conditions and time points in samples without hepcidin (Figure 7A). However, in the presence of hepcidin (Figure 7B), longitudinal relaxation rate significantly increased upon iron supplementation (−Fe versus +Fe, *p* < 0.01) and remained elevated up to four hours after the removal of iron supplementation. Comparison of hepcidin treatment using two-way ANOVA (Figure 7C) showed a significant decrease in R_1_ (*p* < 0.05) at 1 h-Fe in the presence of hepcidin. Regardless, there was no significant change in cellular iron content with and without hepcidin at 1 h-Fe.

### 2.7. Correlation between Cellular Iron Content and Transverse Relaxation

To understand the correlation between cellular iron content (the independent variable) and transverse relaxation rate (the dependent variable), Pearson’s correlation test was performed. There was a moderate positive correlation (Figure 8A, r = 0.62, *p* < 0.01) between cellular iron content and the total transverse relaxation rate, R_2_*; however, no correlation was obtained between cellular iron content and the irreversible R_2_ component of transverse relaxation rate. In THP-1 monocytes, the moderate positive correlation (Figure 8B, r = 0.61, *p* < 0.01) between cellular iron content and transverse relaxation rests with the reversible R_2_’ component. The lines of best fit were determined using a linear regression model and an independent samples t-test indicates that these relationships have similar slopes, between 5.8 and 8.8.

For the samples treated with hepcidin, Pearson’s correlation analysis shows a strong positive correlation between cellular iron content and R_2_* (Figure 9A, r = 0.80, *p* < 0.001). Indeed, after THP-1 monocytes are treated with hepcidin, there is a strong positive correlation between cellular iron content and both R_2_ (Figure 9B, r = 0.71, *p* < 0.01) and R_2_’ (Figure 9C, r = 0.75, *p* < 0.001). Using a linear regression model, the lines of best fit were determined. An independent samples t-test indicates that the slopes of these lines are similar for all transverse relaxation rates. Moreover, comparison of the slopes between hepcidin-treated and untreated samples revealed there were no significant changes in the slope when samples were exposed to hepcidin.

## 3. Discussion

Monocytes are the most abundant cell type during an inflammatory response [2]. As such, different monocyte subpopulations reflecting pro- and anti-inflammatory stages may potentially serve as biomarkers for monitoring inflammation. In our study, we used the human THP-1 monocytic cell line to monitor hepcidin-FPN interactions, which occur in only a few cell types and may provide a relatively unique biomarker for molecular imaging.

### 3.1. Iron Export in Monocytes

Hepcidin is a polypeptide hormone secreted in response to changes in systemic iron and pro-inflammatory cytokines such as interleukin 6 (IL-6). Hepcidin activity downregulates the iron export protein, FPN. By culturing monocytes in the presence and absence of hepcidin (200 ng/mL), while treating the cells with extracellular iron supplementation (25 µM ferric nitrate), we investigated the influence of iron export on MR relaxation rates. In addition, we measured the cellular iron content and correlated it with MR relaxation rates to better understand the relationship between these measures.

As the main mammalian iron export protein, FPN recycles intracellular iron back to the plasma. FPN is only expressed by certain cells, predominantly hepatocytes, enterocytes, macrophages and their precursors—the monocytes [13]. In response to inflammation, monocytes phagocytose damaged red blood cells and then release the iron recovered from heme back into plasma, predominantly for the synthesis of new red blood cells [14]. As the expression of FPN is integral to this process, in our study, we confirmed that THP-1 monocytes express the iron export protein FPN. Relative to the housekeeping protein, GAPDH, the expression of FPN is abundant (ratio~1). These data are supported by literature related to FPN expression in freshly isolated human blood monocytes and THP-1 monocytes [15,16]. Our observations of FPN expression in the presence of extracellular iron supplementation (+Fe samples) are consistent with a study showing an increase in FPN mRNA levels in THP-1 cells under iron-supplemented conditions [17]. Given the relatively equal levels of FPN protein +/− Fe, these findings further suggest that FPN turnover, through both transcriptional and post-translational mechanisms, may be active in monocytes in the presence of an extracellular iron supplement. FPN protein expression was downregulated within 1 to 2 h after the removal of iron supplementation, consistent with post-translational regulation of FPN and consistent with the reported production of hepcidin by THP-1 monocytes, which leads to FPN ubiquitination [14,16]. This hypothesis implicates hepcidin autocrine/paracrine activity, which we documented upon withdrawal of iron supplementation (in the absence of exogenous hepcidin). We note that no further degradation of FPN by exogenously added hepcidin was obtained at 1 h-Fe and 2 h-Fe, as would be expected for autocrine/paracrine regulation.

Interestingly, FPN returns to baseline levels (high expression) by 4 h-Fe only to drop again at 24 h-Fe, indicating a biphasic response to (the presumed) monocyte self-regulation by hepcidin. The short-lived nature of the active form of hepcidin (hepcidin-25) permits fine-tuned control of FPN, and therefore, the iron export function of monocytes.

### 3.2. Hepcidin Regulation of Monocytes

When pro-inflammatory signaling through IL-6 promotes inflammation, the hormone hepcidin is produced by the liver [11,18]. Through downregulation of FPN in monocytes and macrophages, hepcidin reduces the extracellular iron availability at the site of inflammation. Consequently, hepcidin helps fight infection since iron is a critical co-factor for many microbes [19,20]. In our study, we detected downregulation of ferroportin expression in THP-1 monocytes in response to exogenous hepcidin treatment whenever monocyte self-regulation did not predominate: at baseline (−Fe), in the presence of continuous iron supplementation (mimicking hemorrhage, +Fe), and at 4 h-Fe. Hence, monocytes appear to be subject to endocrine, paracrine and autocrine regulation by hepcidin [16].

Over the years, clinicians have been struggling to distinguish between pro-inflammatory and anti-inflammatory stages after acute myocardial infarction (AMI) to prevent unwanted tissue remodeling leading to heart failure [21,22]. Since monocytes along with hormones such as hepcidin travel to the infarcted myocardium through systemic circulation post-AMI, we expected that pro-inflammatory signaling through hepcidin may cause increased iron retention in monocytes and facilitate their detection using MRI. In the absence of exogenous hepcidin (low/no endocrine signaling), total cellular iron content is not significantly altered by changes in extracellular iron, pointing to the efficiency of autocrine/paracrine hepcidin-mediated regulation of iron homeostasis in an iron-exporting cell type. While this form of monocyte iron regulation does not influence R_1_, it is transiently detected by R_2_* and specifically the reversible R_2_’ component. Indeed, under these conditions, there was no correlation between iron and R_2_; however, there was a moderate correlation between iron and both R_2_* and R_2_’. To the extent that R_2_ represents protein-bound iron, it appears that autocrine/paracrine regulation of FPN largely influences the unbound iron fraction that modulates the R_2_’ signal.

Our examination of pseudo-endocrine regulation of FPN, by exogenously added hepcidin, resulted in discrete changes not only in total cellular iron content, but also in all three transverse relaxation rates. Exogenous hepcidin significantly influenced R_2_, which reflected the same pattern of changes documented for R_2_*. These MR signals were influenced by both long-term (continuous iron supplementation) and short-term (within 24 h of iron supplement withdrawal) changes in extracellular iron. By comparison, R_2_’ mainly displayed changes over shorter time scales, within 24 h of removing iron supplementation. Moreover, in the presence of exogenous hepcidin, the correlations between total cellular iron content and transverse relaxation rates were all notable. All transverse relaxation rates displayed a significantly strong correlation to iron. We note that both longitudinal and transverse relaxation rates identify a significant effect of hepcidin in the one-hour time frame (1 h-Fe), possibly reflecting the short half-life of hepcidin and the transient nature of its activity. The changes in iron biochemistry that underlie these differential MR responses to autocrine/paracrine and endocrine hepcidin regulation of monocytes are considered below.

### 3.3. Monocyte Iron Biochemistry

Mammalian cells predominantly take up ferric iron (Fe^3+^) through transferrin-transferrin receptor (TfRc) interactions and store it as a biomineral in ferritin, in the ferrous state (Fe^2+^). In humans, there is a positive correlation between serum ferritin and hepcidin indicating they work together to promote cellular iron storage [23]. The latter study also reported an increase in serum hepcidin expression during inflammation. It is not clear whether the stimulus, concentration and/or duration of the hepcidin signal influences the extent of monocyte responses to hepcidin activity. Our MR data suggest that iron-stimulated changes in FPN expression, resulting in autocrine/paracrine hepcidin activity, are distinct from endocrine hepcidin activity.

In our cell model, in the presence of extracellular iron supplementation (+Fe), there was no significant change in cellular iron content compared to the untreated control. Although another report suggested this may be due to an increase in FPN expression in iron-supplemented cells [24], we did not detect any change in iron export protein. While we have not ruled out the possibility that low TfRc expression limits iron uptake, following both hepcidin and extracellular iron treatment we observed a significant increase in cellular iron content compared to unsupplemented controls (−Fe). This finding is consistent with a key role for iron export in the maintenance of monocyte iron homeostasis. As reported for hepatoma and THP-1 co-cultures [10], we showed that multi-potent P19 cells also secrete hepcidin activity that reduces FPN in THP-1 cells.

Another hypothesis is that hepcidin balances the fraction of iron stored in ferritin versus the labile iron pool (LIP), attenuating the net change in total cellular iron content. Since the form of iron cannot be determined through ICP-MS measurements, we have no indication of relative levels of iron in ferritin or LIP. This may be important to differentiate in the future as studies have shown increased ferritin mRNA expression due to hepcidin [17] as well as a positive correlation between monocyte LIP and hepcidin expression [25].

Where changes in MRI signal in the presence and absence of hepcidin are not corroborated by the quantification of iron, we speculate that differences in intracellular iron redox status may contribute to changes in transverse relaxation. Dietrich et al. (2017) show a significant increase in R_2_* of ferric (Fe^3+^) compared to ferrous (Fe^2+^) ions [26]. In addition, House et al. (2007) emphasize that an increase in iron stored as ferritin may lead to an increase in R_2_, partially consistent with our results [27]. The difference in correlation times (defined as the time required to rotate by approximately 1 radian) of dipolar interactions between the redox state(s) of iron and water protons contributes to changes in relaxation rates [26,28,29].

In large-animal and human imaging, MRI provides the best resolution and depth of penetration for soft tissues, with a voxel size on the order of 1 mm^3^ [30]. However, the sensitivity of MRI is relatively low, requiring micromolar concentrations of contrast agent, in the case of iron nanoparticles. For molecular imaging with MRI, superparamagnetic iron oxide (SPIO) nanoparticles have been widely used to label cells [31]. These exogenous contrast agents produce detectable changes in MRI signals that are nevertheless lost upon degradation and mitosis. By comparison, gene-based iron labelling provides long-term contrast for the non-invasive detection of cell status. However, apart from magnetotactic bacteria, few cells can achieve the same degree of iron labelling that SPIO delivers. To address this, some success has been obtained by genetically programming iron contrast in mammalian cells [32]. There is a strong linear relationship between total cellular iron content and transverse relaxation rates. Moreover, in iron-exporting cell types such as undifferentiated P19 cells, the slope of this linear correlation increases in the presence of hepcidin [12], by approximately threefold for R_2_* (from 10.1 to 35.4 s^−1^/µg Fe/mg protein; *p* < 0.05) and by approximately fivefold for R_2_ (from 5.25 to 27.8 s^−1^/µg Fe/mg protein; *p* < 0.01). Thus, the regulation of iron export may be effective at improving cellular iron contrast for molecular MRI in suitable target cells, including monocytes and macrophages.

An appreciation of how well monocyte filled volumes may be detected in vivo by endogenous changes in iron metabolism is still not clear. However, hepcidin strengthens the relationship between transverse relaxation rates and cellular iron content in both P19 cells [12] and THP-1 monocytes, reproducibly showing the greatest changes in the R_2_ component (from no correlation to a slope of 7.81 s^−1^/µg Fe/mg protein in THP-1 cells treated with hepcidin; r = 0.71). For clinically relevant imaging, the form of cellular iron, and its relationship to R_2_* and R_2_ transverse relaxation rates, should be investigated. This information may clarify how best to use these MR parameters for longitudinally tracking the course of endocrine hepcidin activity. In addition, while the MR measures presented here were acquired from cell phantoms by assessing volumes fully occupied by monocytes (reflecting proliferation), no attempt was made to characterize this population of cells relative to cluster of differentiation (CD) antigens (e.g., CD14, CD16) before and after hepcidin treatment [1,2,3]. Potentially, the best biomarkers of hepcidin regulated pathways involve a subset of monocytic cells. The present in vitro study, documenting changes in MR relaxation rates that are driven by hepcidin regulation at the cellular level, is a step toward understanding and exploiting changes in tissue iron homeostasis that will improve inflammation imaging at the time of injury and during the healing process.

## 4. Materials and Methods

### 4.1. Reagents

Unless otherwise indicated, the reagents were purchased from Thermo Fisher Scientific (Mississauga, ON, Canada) and from Sigma-Aldrich (Oakville, ON, Canada).

### 4.2. THP-1 Monocyte Model

Human THP-1 monocytes, derived from the peripheral blood of a male acute monocytic leukemia patient (ATCC TIB-202), were cultured as a cell suspension in RPMI-1640 medium/10% fetal bovine serum/4 U/mL penicillin/4 µg/mL streptomycin/50 µM 2-mercaptoethanol. Cells were incubated at 37 °C in a 5% CO_2_/air mixture. Cultures were maintained between 2–8 × 10^5^ cells/mL based on cell counts determined through hemocytometry.

### 4.3. Iron Supplementation

THP-1 cells were resuspended at a concentration of 2–4 × 10^5^ cells/mL and cultured in the absence (−Fe) or presence (+Fe) of iron-supplemented medium containing 25 µM ferric nitrate for 7 days (Figure 10A). Iron-supplemented cells were then returned to non-supplemented medium and cultured for an additional 1 (1 h-Fe), 2 (2 h-Fe), 4 (4 h-Fe), and 24 (24 h-Fe) hours (Figure 10B). At each time point, cells were collected in 850 µL radioimmunoprecipitation assay buffer (RIPA; 10 mM Tris-HCl pH 7.5/140 mM NaCl/1% NP-40/1% sodium deoxycholate/0.1% sodium dodecyl sulfate [SDS]) containing 150 μL Complete Mini protease inhibitor cocktail, Roche Diagnostic Systems (Laval, QC, Canada), for protein analysis.

### 4.4. Hepcidin Treatment

To examine the response of THP-1 monocytes to hepcidin, cells were cultured in the presence and absence of iron supplement, as described above, and in the presence of 200 ng/mL hepcidin for up to 24 h (Figure 10C,D). Following hepcidin treatment, cells were prepared for protein analysis, elemental iron analysis or MRI.

For MRI experiments, THP-1 monocytes subjected to different iron treatment conditions in the presence and absence of hepcidin were collected intact by centrifugation at 300× *g* for 5 min at 15 °C. After centrifugation, the medium was removed and cells were washed three times with phosphate buffered saline pH 7.4 (PBS). Cells were then centrifuged at 300× *g* for 5 min at 15 °C in custom-made Ultem wells (inner diameter: 4 mm; height: 10 mm, Lawson Imaging Prototype Lab). Once the cells were loaded into the Ultem wells, MRI relaxation rates were measured as described below.

### 4.5. THP-1 Cell Culture with Conditioned Medium

Mouse P19 embryonic carcinoma cells were cultured in alpha-minimum essential medium (α-MEM)/10% FBS/4 U/mL penicillin/4 µg/mL streptomycin supplemented with 25 μM ferric nitrate for 5–7 days as previously described [12]. To stimulate hepcidin secretion, iron supplement was removed, and cells were further cultured in non-supplemented α-MEM for an additional 24 h. This stimulated medium, presumed to be enriched with hepcidin, was centrifuged at 1000× *g* for 2 min to remove any cells prior to diluting supernatants 1/2 with RPMI 1640 medium for culture with THP-1 monocytes. After 24 h, THP-1 samples were harvested. Protein concentration was determined by the bicinchoninic acid (BCA) assay. THP-1 cells cultured in RPMI 1640 alone served as a control. Non-stimulated medium consisted of P19 cell supernatant not exposed to iron supplement. Relative changes in FPN and glyceraldehyde-3-phosphate dehydrogenase (GAPDH) protein levels were determined from western blots using primary rabbit antibodies to FPN and GAPDH, respectively, as described below.

### 4.6. Protein Preparation and BCA Assay

For western blots and elemental iron analysis, cells cultured under different conditions were washed 3 times in PBS, centrifuging at 300× *g* for 5 min at 15 °C to remove the supernatant. The final cell pellet was collected in 850 µL RIPA buffer with 150 μL Complete Mini protease inhibitor cocktail. Samples were placed on ice and complete cell lysis was achieved by sonicating three rounds, each for 12 s using a Sonic Dismembrator 500, Fischer Scientific (Pittsburg, PA, USA). Protein concentrations were determined using the BCA assay with bovine serum albumin (BSA) as the protein standard [33].

### 4.7. Western Blot

Samples containing total cellular protein were separated by SDS polyacrylamide gel electrophoresis (SDS-PAGE) according to published procedures [34]. Samples were assessed under reducing conditions using 1 mM dithiothreitol. Each sample contained 20 µg protein and was separated on a 10% running gel.

Separated protein was transferred to a nitrocellulose blot (iBlot Gel Transfer Stacks) following the manufacturer’s protocol and published procedures [35]. To block nonspecific binding, the blot was incubated in 6% BSA/10 mM Tris HCl pH 7.4/0.9% NaCl (Tris buffered saline, TBS)/0.02% sodium azide (TBSA) for a minimum of 2 h at room temperature. To check for expression of FPN protein, the blot was incubated overnight at room temperature in 1:2000 rabbit anti-FPN/3%BSA/TBSA. After incubation with primary antibody, blots were washed for 30 min with 4 changes of TBS/0.1% Tween 20 (TBST) and then incubated for 2 h at room temperature in 1:10,000 horseradish peroxidase (HRP)-conjugated goat anti-rabbit immunoglobulin (Ig)/1% BSA/TBS. After incubation in secondary antibody, blots were washed in TBST for 30 min with two changes of buffer. Bands were developed using the SuperSignal West Pico Chemiluminescent Substrate, detecting signal with the ChemiDoc^®^ Imaging System, Syngene (Frederick, MD, USA). The molecular weight (M.W.) of FPN is approximately 63 kDa [36].

FPN expression was compared to the GAPDH control. Blots were stripped in solution containing 245 mM β-mercaptoethanol/2% SDS/62.5 mM Tris-HCl (pH 6.8) and reprobed as described above, using 1:2000 rabbit anti-GAPDH as the primary antibody. The M.W. of GAPDH is approximately 37 kDa.

Expression of FPN was analyzed by densitometry using ImageJ software. The signal intensity of FPN was normalized to the signal intensity of GAPDH.

### 4.8. Trace Elemental Iron Analysis

To determine elemental iron content, samples containing 2–4 mg/mL of total cellular protein (described above) were analyzed by inductively-coupled plasma mass spectrometry (ICP-MS; Biotron Analytical Services, Western University). Briefly, samples were digested using nitric acid and heat, then filtered prior to ICP-MS. Total cellular iron content was normalized to the total amount of protein. Each value reported for elemental iron was obtained from the same batch of cells divided between MRI and ICP-MS analyses.

### 4.9. MRI Phantoms

Each sample consisted of approximately 40–50 million cells placed in an Ultem well prior to mounting in a 9 cm, spherical 4% gelatin (porcine type A)/PBS phantom (Figure 11A) [37]. Cell phantoms were scanned at 3 Tesla (3T) on a Biograph mMR, Siemens AG (Erlangen, Germany) using previously developed sequences to acquire relaxation rates [37]. Single echo spin echo and multi-echo gradient echo sequences were applied to obtain R_2_ and R_2_*, respectively. R_2_’ was calculated by subtraction (R_2_* − R_2_) [37]. The following imaging parameters were used for MR image acquisition. For the R_1_ longitudinal relaxation rate, an inversion recovery spin echo sequence (IR) was used. The slice thickness was 3 mm (Figure 11B), matrix size was 128 × 128, Partial Fourier was 6/8, voxel size was 3.0 × 0.9 × 0.9 mm^3^, repetition time (TR) was 4000 ms, flip angle was 90°, inversion times were 22, 200, 500, 1000, 2000 and 3900 ms and echo time (TE) was 13 ms. Total scanning time for the IR acquisition was approximately 39 min. For both single echo spin echo and multi-echo gradient echo sequences, slice thickness was 3 mm, matrix size was 192 × 192 and voxel size was 3.0 × 0.6 × 0.6 mm^3^. For the single echo spin echo sequence, the quantity TR-TE was held fixed [38] at 2000 ms. Values of TE were 13, 20, 25, 30, 40, 60, 80, 100, 150, 200 ms; flip angle was 90°; and scanning time was approximately 67 min. For multi-echo gradient echo, TE values were 6.12, 14.64, 23.16, 31.68, 40.2, 50, 60, 70, 79.9 ms; TR was 2000 ms; number of signal averages = 4; flip angle was 60° and scanning time was approximately 25 min.

Transverse relaxation rates (R_2_* and R_2_) were measured using software developed in-house using Matlab 7.9.0 (R2010b). This software was used to determine the region of interest (ROI) when measuring R_2_* and R_2_ signals. The ROI were manually contoured and included 21 voxels within the sample, avoiding the wall of the well as visualized in axial cross section. Relaxation rates were calculated using the average signal intensity and least-squares curve fitting. Relaxation rates were reported as the mean +/− standard error of the mean (SEM) using GraphPad Prism software, version 8.0.0.

### 4.10. Statistical Analysis

Statistical analyses were performed using SPSS, version 25. MR relaxation rates obtained for treatment time points, reflecting changes in extracellular iron supplementation or hepcidin treatment, were analyzed using one-way analysis of variance (ANOVA). Significant differences were defined by *p* < 0.05. To test the effect of the combination of hepcidin and iron supplementation on MR relaxation rates, a two-way ANOVA was conducted to determine significant differences. Pearson’s correlations were assessed to determine the relationship between total cellular iron content and MR relaxation rate. A linear regression model was applied using cellular iron content as the independent variable and MR relaxation rate as the dependent variable to obtain the line of best fit.

For analyses related to conditioned medium, ANOVA and Tukey’s honestly significant difference (HSD) test were performed with RStudio software.

## 5. Conclusions

Differences in cellular iron-handling mechanisms may allow some cell types to be tracked using MRI [39]. For example, monocytes are one of the most abundant immune cell types in chronic inflammatory conditions like atherosclerosis. In this context, hepcidin-mediated changes in iron retention may allow us to track the activity of these cells. Toward this goal, we examined human THP-1 monocytes and the effect of hepcidin on FPN protein expression, cellular iron content, and MR relaxation rates.

Since there was little or no change in FPN expression upon iron supplementation (+Fe), cellular iron content and MR relaxation rates remained unchanged compared to non-supplemented monocytes (−Fe). Following the withdrawal of iron supplement, cellular iron content remained constant; although, R_2_* and R_2_’ significantly increased. In general, the addition of hepcidin lead to downregulation of FPN and a significant increase in cellular iron content. This finding was reflected in a significant increase in both longitudinal and transverse relaxation rates. Moreover, while iron content and R_2_* were positively correlated in the absence of hepcidin, this moderate correlation was strengthened by the addition of hepcidin. These results demonstrate the effect of hepcidin on THP-1 monocyte iron regulation and the degree to which cellular iron content and regulation of iron export activity affect MRI measures. Overall, results from our study suggest that hepcidin-mediated changes in monocyte iron handling could potentially be tracked using MRI and may be exploited for monitoring inflammatory processes in vivo.

## Figures and Tables

**Figure 1 ijms-24-04036-f001:**
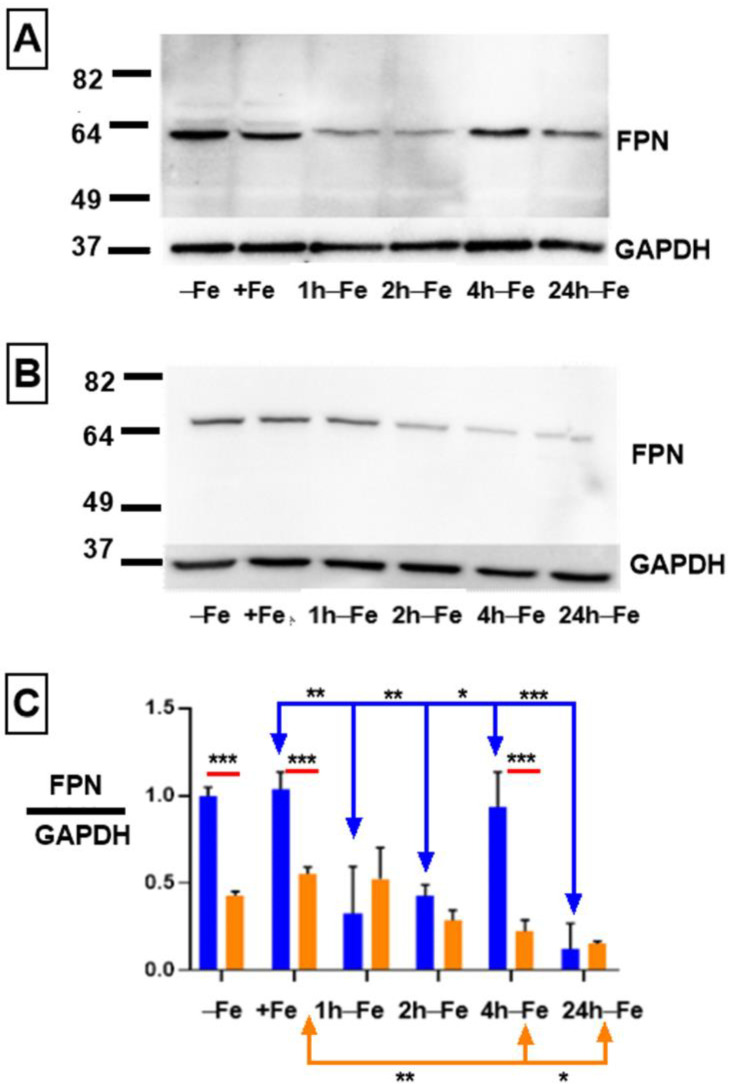
Regulation of FPN expression in THP-1 monocytes by extracellular iron and hepcidin. THP-1 cells were cultured for 7 days in the absence (−Fe) or presence (+Fe) of iron-supplemented medium containing 25 µM ferric nitrate. Cells were either harvested immediately or after the withdrawal of iron supplement and culture for an additional 1 (1 h-Fe), 2 (2 h-Fe), 4 (4 h-Fe) and 24 (24 h-Fe) hours. To examine FPN regulation, −Fe and +Fe samples were grown in the presence of 200 ng/mL hepcidin for the last 24 h of culture while hepcidin was added to all other samples after the removal of iron supplement. Representative western blots show the change in FPN expression in the absence (**A**) and presence (**B**) of hepcidin. Molecular weight standards are indicated on the left while GAPDH provided a loading control. Densitometry (**C**) indicates relative level of FPN expression normalized to GAPDH (n = 3 independent experiments, * *p* < 0.05, ** *p* < 0.01, *** *p* < 0.001). Blue lines indicate significant differences in the ratio of FPN/GAPDH resulting from changes in extracellular iron (blue bars). Orange lines indicate significant differences in FPN/GAPDH resulting from the addition of hepcidin (oranges bars). Red lines compare significant differences in FPN/GAPDH between the two treatment groups (+/− hepcidin).

**Figure 2 ijms-24-04036-f002:**
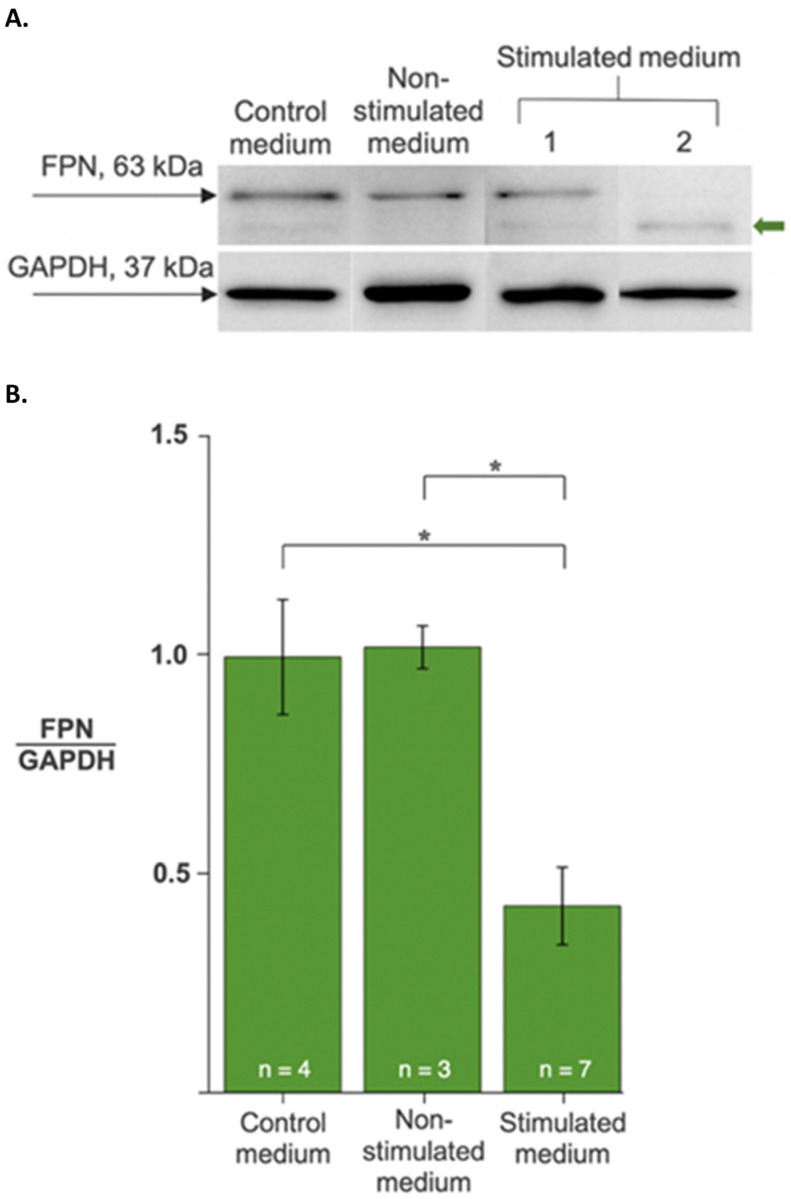
Regulation of ferroportin levels in monocytes by secreted hepcidin activity. THP-1 cells were cultured in the absence (−Fe, control medium) or presence of P19 cell-conditioned medium, either non-stimulated or stimulated to secrete hepcidin. (**A**) FPN (upper panel) and GAPDH (lower panel) protein levels were determined from western blots. The green arrow indicates varying degrees of FPN degradation in representative samples. (**B**) Changes in FPN protein levels are expressed as the ratio of FPN to GAPDH. Data are normalized to the control medium treatment. Samples treated with stimulated medium showed an approximately 2-fold decrease in FPN relative to controls. Values are the mean ± SEM. *, *p* < 0.01.

**Figure 3 ijms-24-04036-f003:**
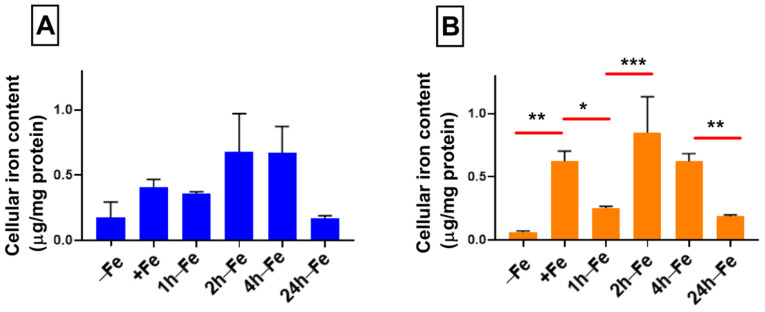
Influence of extracellular iron and hepcidin on intracellular iron content. To examine the effect of changes in extracellular iron (**A**), THP-1 cells were cultured for 7 days in the absence (−Fe) or presence (+Fe) of iron-supplemented medium containing 25 µM ferric nitrate. Cells were either harvested immediately (−Fe and +Fe) or after removal of iron supplement and culture for an additional 1 (1 h-Fe), 2 (2 h-Fe), 4 (4 h-Fe) or 24 (24 h-Fe) hours in non-supplemented medium. To examine the regulation of iron export (**B**), −Fe and +Fe samples were grown in the presence of 200 ng/mL hepcidin for the last 24 h of culture while hepcidin was added to all other samples after the removal of iron supplement. Total cellular iron content was assessed by ICP-MS and was normalized to protein concentration. In response to changes in extracellular iron, the total intracellular iron content ranged between 0.175 and 0.679 µg/mg protein but was not significantly different between samples (n = 3–4). However, in the presence of hepcidin, iron-supplemented cells (+Fe) retained significantly more cellular iron than non-supplemented cells (−Fe) but returned to baseline values within 1 h after the withdrawal of iron supplement. A biphasic response to the addition of hepcidin was observed between 2 and 24 h after the withdrawal of iron supplement (2 h-Fe to 24 h-Fe; n = 3–4). * *p* < 0.05, ** *p* < 0.01, *** *p* < 0.001.

**Figure 4 ijms-24-04036-f004:**
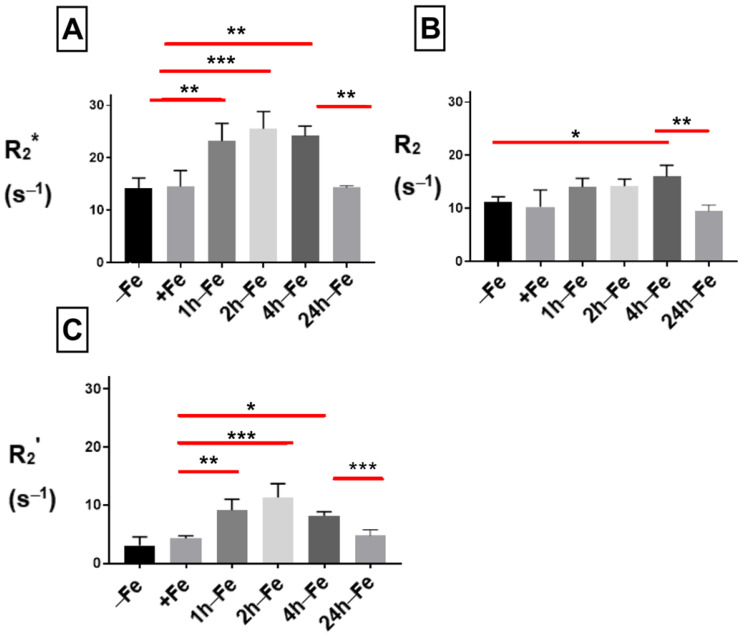
Influence of extracellular iron on transverse relaxation rates in human monocytes. To examine the influence of extracellular iron, THP-1 cells were cultured in the absence (−Fe) or in the presence (+Fe) of iron-supplemented medium containing 25 µM ferric nitrate for 7 days. Cells were then harvested and scanned either immediately (−Fe and +Fe) or cultured for an additional 1 (1 h-Fe), 2 (2 h-Fe), 4 (4 h-Fe) and 24 (24 h-Fe) hours after removal of extracellular iron supplement. One-way ANOVA indicates significant changes between samples subjected to iron treatments. (**A**) Regardless of iron supplementation (+/− Fe), human THP-1 cells displayed relatively high transverse relaxation rates (R_2_*). However, within an hour of iron supplement withdrawal, R_2_* significantly increased and remained elevated up to 4 h before returning to baseline. (**B**) The irreversible component of transverse relaxation (R_2_) increased modestly at 4 h-Fe before returning to baseline. (**C**) Like R_2_*, the reversible component (R_2_’ = R_2_* − R_2_) increased significantly upon withdrawal of iron supplementation and remained elevated up to 4 h before returning to baseline (n = 3–4, * *p* < 0.05, ** *p* < 0.01, *** *p* < 0.001).

**Figure 5 ijms-24-04036-f005:**
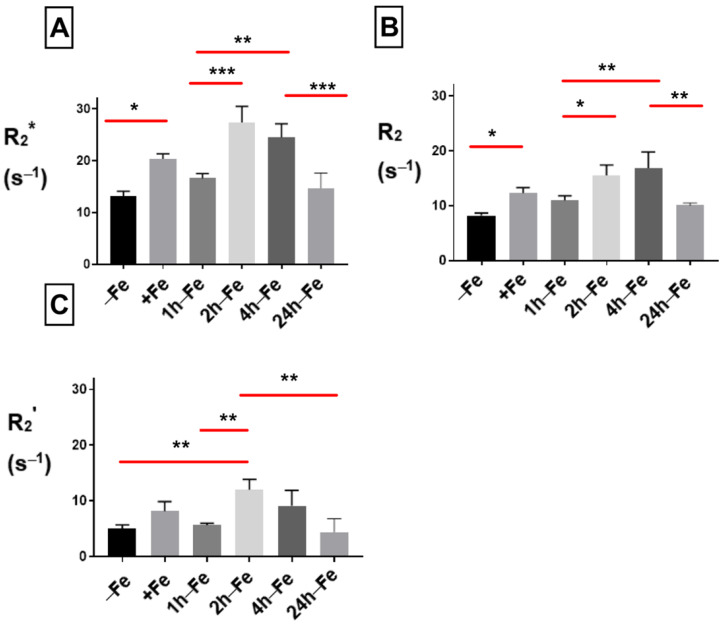
Influence of extracellular iron and hepcidin on transverse relaxation rates in human monocytes. To examine the influence of extracellular iron and hepcidin, THP-1 cells were cultured in the absence (−Fe) or in the presence of (+Fe) iron-supplemented medium containing 25 µM ferric nitrate for 7 days. Cells were harvested either immediately or 1 (1 h-Fe), 2 (2 h-Fe), 4 (4 h-Fe) and 24 (24 h-Fe) hours after removal of extracellular iron supplementation. In addition, −Fe and +Fe samples were grown in the presence of 200 ng/mL hepcidin for the last 24 h of culture while hepcidin was added to all other samples after removal of iron supplementation. One-way ANOVA indicates significant changes between samples subjected to iron and hepcidin treatments. (**A**) In the presence of iron supplementation (+Fe), human THP-1 cells display a significant increase in the total transverse relaxation rate (R_2_*). For the first hour after iron supplement withdrawal, R_2_* remains relatively constant. (**B**) The irreversible component (R_2_) also increased significantly in the presence of iron. (**C**) The reversible component (R_2_’) was only significantly increased two hours after iron supplementation withdrawal. n = 3–4, * *p* < 0.05, ** *p* < 0.01, *** *p* < 0.001.

**Figure 6 ijms-24-04036-f006:**
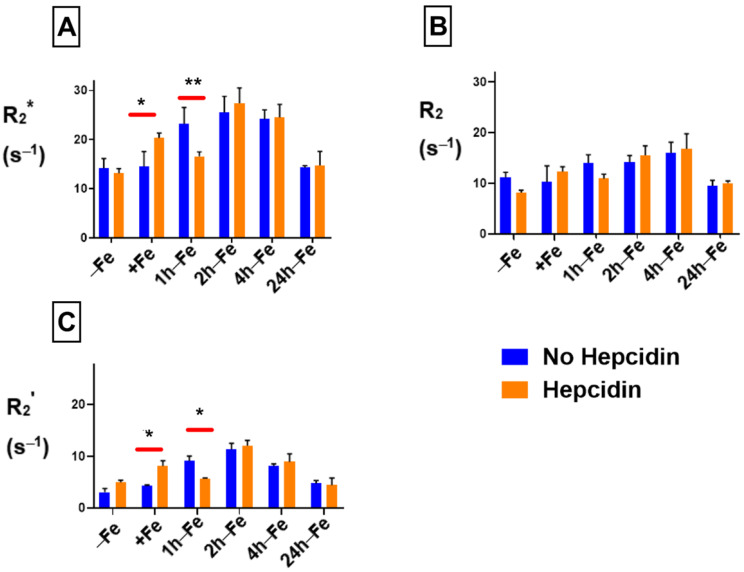
Influence of extracellular iron and hepcidin on transverse relaxation rates in human monocytes. To examine the influence of extracellular iron and hepcidin, THP-1 cells were cultured in the absence (−Fe) or in the presence of (+Fe) iron-supplemented medium containing 25 µM ferric nitrate for 7 days. These samples were treated with 200 ng/mL hepcidin for the last 24 h of culturing. Cells were harvested either immediately or 1 (1 h-Fe), 2 (2 h-Fe), 4 (4 h-Fe) and 24 (24 h-Fe) hours after removal of extracellular iron supplementation. Furthermore, these samples were treated with 200 ng/mL of hepcidin for up to 24 h before the harvest. Two-way ANOVA indicates significant changes between samples treated in the presence and absence of hepcidin. Both R_2_* (**A**) and R_2_’ (**C**) relaxation rates increase significantly in the presence of iron supplement and hepcidin, with a significant decrease within an hour of iron supplementation withdrawal. The irreversible component R_2_ (**B**) is not significantly influenced by hepcidin. n = 3–4, * *p* < 0.05, ** *p* < 0.01.

**Figure 7 ijms-24-04036-f007:**
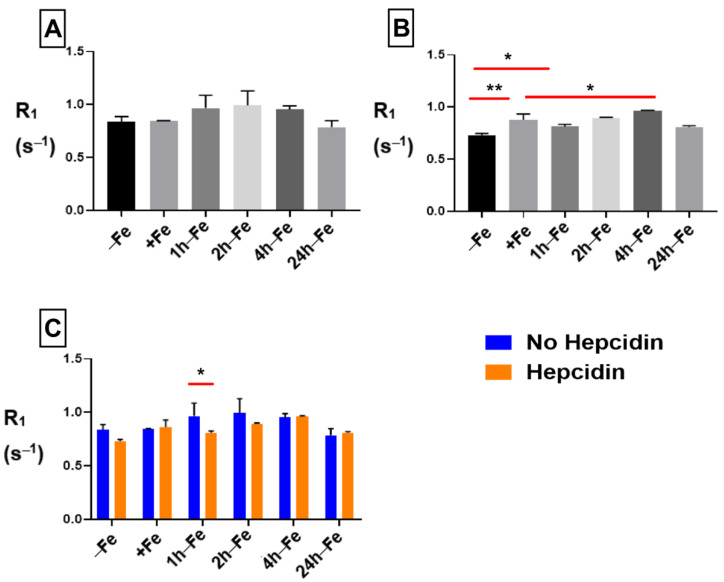
Influence of extracellular iron and hepcidin on longitudinal relaxation rates. To examine the influence of extracellular iron and hepcidin, THP-1 cells were cultured in the absence (−Fe) or presence (+Fe) of iron-supplemented medium containing 25 µM ferric nitrate for 7 days. These samples were treated with 200 ng/mL hepcidin for the last 24 h of culture. Cells were harvested either immediately or 1 (1 h-Fe), 2 (2 h-Fe), 4 (4 h-Fe) and 24 (24 h-Fe) hours after removal of extracellular iron supplement. Furthermore, these samples were treated with 200 ng/mL of hepcidin for up to 24 h before the harvest. One-way ANOVA indicates that (**A**), in the absence of hepcidin, R_1_ remains relatively constant across all samples; and (**B**), in the presence of hepcidin, R_1_ significantly increased following iron supplementation (+Fe) and remained elevated up to 4 h after the withdrawal of iron supplementation. Two-way ANOVA indicates that (**C**), in the presence of hepcidin, there was a significant decrease in R_1_ at 1 h-Fe. n = 3, * *p* < 0.05, ** *p* < 0.01.

**Figure 8 ijms-24-04036-f008:**
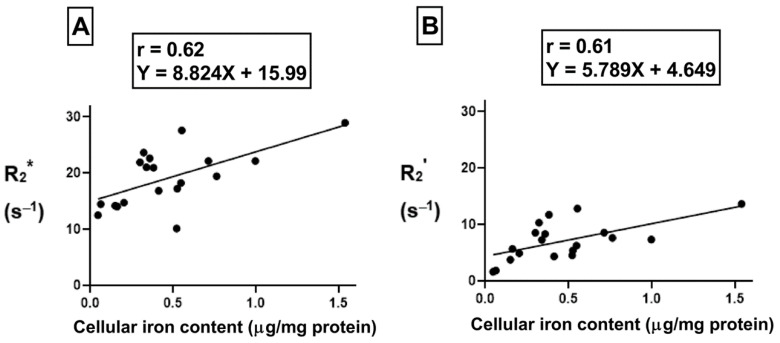
Influence of changes in extracellular iron on the correlation between cellular iron content and MR transverse relaxation rates. To examine the influence of extracellular iron, THP-1 cells were cultured in the absence (−Fe) or in the presence (+Fe) of iron-supplemented medium containing 25 µM ferric nitrate for 7 days. Cells were then harvested and scanned either immediately (−Fe and +Fe) or cultured an additional 1 (1 h-Fe), 2 (2 h-Fe), 4 (4 h-Fe) and 24 (24 h-Fe) hours after removal of extracellular iron supplementation. Total cellular iron content was assessed by ICP-MS and was normalized to the amount of protein. There is a moderate positive correlation between cellular iron content and both R_2_* (**A**) and R_2_’ (**B**). For both graphs, n = 19 and *p* < 0.01.

**Figure 9 ijms-24-04036-f009:**
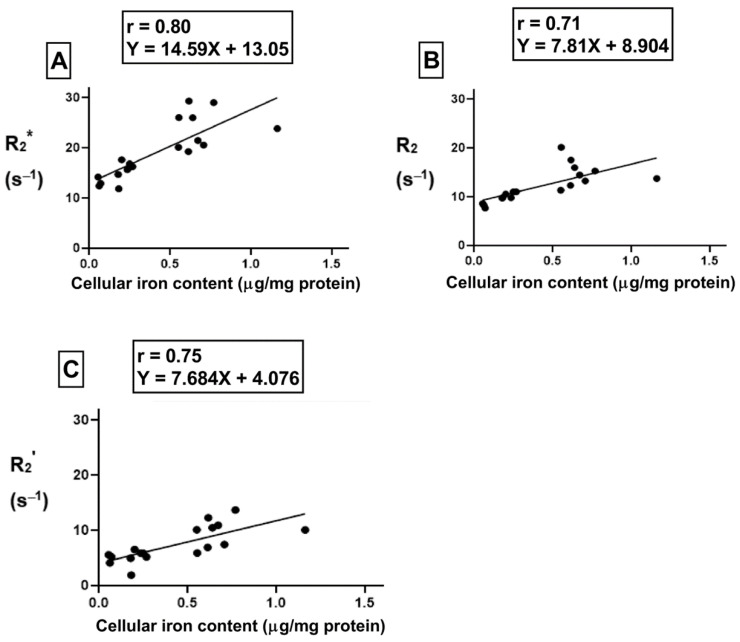
Influence of hepcidin on the correlation between cellular iron content and transverse relaxation rates. To examine the influence of extracellular iron and hepcidin, THP-1 cells were cultured in the absence (−Fe) or in the presence of (+Fe) iron-supplemented medium containing 25 µM ferric nitrate for 7 days. These samples were treated with 200 ng/mL hepcidin for the last 24 h of culture. Cells were harvested either immediately or 1 (1 h-Fe), 2 (2 h-Fe), 4 (4 h-Fe) and 24 (24 h-Fe) hours after removal of extracellular iron supplement. These samples were also treated with 200 ng/mL of hepcidin for up to 24 h before harvest. Total cellular iron content was assessed by ICP-MS and normalized to the amount of protein. There is a strong correlation between cellular iron content and R_2_* (*p* < 0.001, **A**), R_2_ (*p* < 0.01, **B**) and R_2_’ (*p* < 0.001, **C**). n = 18.

**Figure 10 ijms-24-04036-f010:**
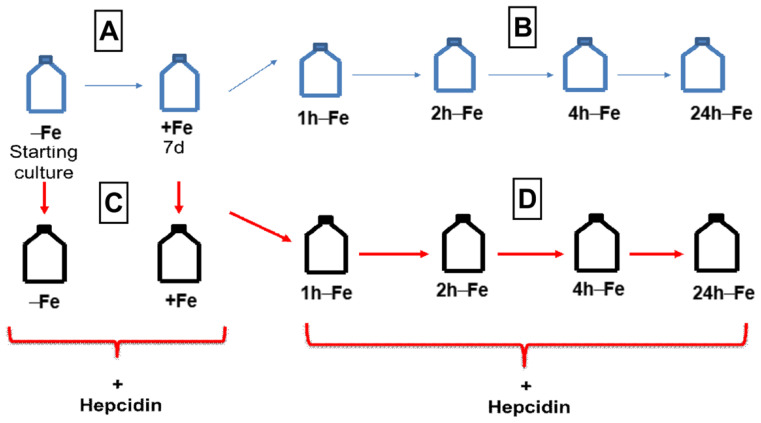
THP-1 cell treatments for western blot and MRI. Cells were cultured in the absence (−Fe) or presence (+Fe) of iron supplementation (25 µM ferric nitrate/medium) for 7 days (**A**) and then harvested either immediately (−Fe and +Fe) or at 1 (1 h-Fe), 2 (2 h-Fe), 4 (4 h-Fe) and 24 (24 h-Fe) hours after removal of extracellular iron supplement (**B**). Similarly, cultured cells were treated with 200 ng/mL hepcidin for up to 24 h of culturing prior to harvest (**C**,**D**).

**Figure 11 ijms-24-04036-f011:**
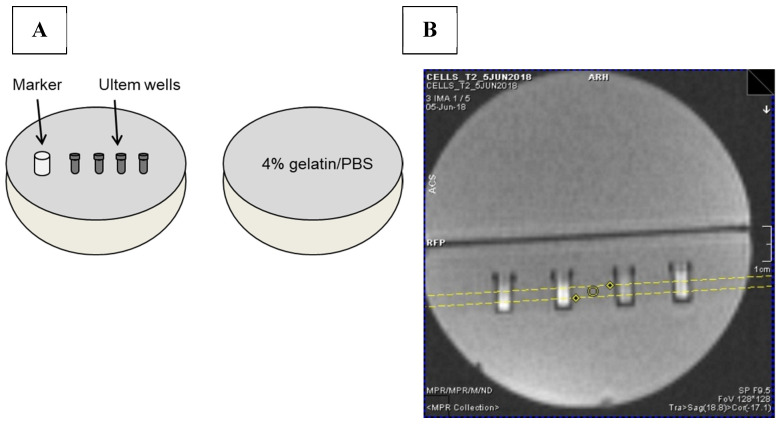
MRI gelatin phantom and slice localization. (**A**) Cells in Ultem wells were mounted in one hemisphere of a 9 cm spherical phantom and overlaid with 4% gelatin/PBS. In the final assembly, this hemisphere was secured to a gelatin-only hemisphere to give a 9 cm sphere overall. A plastic marker was used as an indicator of sample layout [37]. (**B**) To acquire MR images, the cell phantom was placed in a knee coil. Yellow lines in the locator image indicate the 3 mm slice thickness through the samples.

## Data Availability

The data presented in this study are openly available in Scholarship@Western at https://ir.lib.uwo.ca/etd/6390 (accessed on 30 August 2019).

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
