# Peer review of "Monocyte MRI Relaxation Rates Are Regulated by Extracellular Iron and Hepcidin"

_ijms, 2023, doi:10.3390/ijms24044036_

Round 1
Reviewer 1 Report
Title: Monocyte MRI relaxation rates are regulated by extracellular iron and hepcidin
In this study, the author investigated hepcidin-mediated changes in monocyte iron regulation influence both cellular iron content and MRI relaxation rates. The author has addressed the main question of current research and is relevant in the field. However, I have a few minor comments on this manuscript.
Such as
- The author has used a medium supplemented with 10% fetal bovine serum. It has a considerable amount of iron. The present study is mainly focused on iron supplement presence and absence or different forms. So did the author perform any without FBS in the present study?
- The author performed a Cell phantom with 40-50 million cells. Did the author perform a different cell or iron phantom concentration and changes directly proportional to MRI relaxation? It will help to identify how much cellular iron can change in MRI relaxation
- Did the author observe any cellular inflammatory changes in different treatments during this study?
- The author should more details in Trace elemental iron analysis such as preparation and conditions.
Reviewer 2 Report
The topic of the study is interesting for MRI researchers and gave an opportunity to use the suggested approach for tracking particles using MRI. The manuscript reads well and is fluent. I have a couple of points to highlight for the benefit of the readers.
1. In the abstract section, the background was described in great detail yet results are limited without supporting quantitative results. Please consider amending the section by increasing the details of the outcome.
2. Please check the article for misplaced abbreviations. Also, the introduction section is missing the hypothesis. Please revise the section accordingly.
3. Could you please review this statement for clarity. "Also, in the +Fe sample, FPN expression indicates a significant decrease (+Fe blue versus +Fe orange, p < 0.001)."
4. Please provide more quantitative results and metrics.
5. Could you please compare the findings with previous study results? Please mention the limitations of the study.
Round 2
Reviewer 2 Report
Thank you for revising the manuscript according to the reviewers’ comments. The amendment of the manuscript improved the quality of the presentation. However, I have only minor comments for the authors.
1. Please clarify the hypothesis statement. In the recent form, it’s not quite clearly represented the idea of the authors.
2. When citing the companies, state, and country information should be presented within parentheses.
3. Please share the initial concentration of the cells during culturing.
4. For companies located in the USA, only (the city, and state) should be listed as aligned with the literature.
5. In the MRI section, the matrix size shouldn’t have a unit, e.g. 128x128. Also, two of FOV, matrix size, and voxel size would be enough.
6. Please clearly describe the imaging timing for the sequences, you can give for the whole set or each sequence.
7. Could you please describe the segmentation approach used for outlining the ROIs?
8. Is there a specific reason for using three different software platforms for the statistical analyses?
9. I kindly suggest updating figure 3C, it doesn’t look well prepared for representing the outcome.
10. Please check for the format to cite articles “Alizadeh et al, 2020” and figures “new Figure”.
